# Mental Health Screening during COVID-19 Pandemic among School Teachers in Malaysia: A Cross-Sectional Study

**Theingi Maung Maung** [1,*], **Sing Ying Tan** [2], **Chai Li Tay** [3], **Mohammed Shahjahan Kabir** [4], **Lubna Shirin** [5] **and Tan Yong Chia** [6]

1. Unit of Community Medicine, Faculty of Medicine, AIMST University, Bedong 08100, Kedah, Malaysia
2. Hospital Raja Permaisuri Bainun, Ministry of Health Malaysia, Ipoh 30450, Perak, Malaysia
3. Simpang Health Clinic, Ministry of Health Malaysia, Simpang, Taiping 34700, Perak, Malaysia
4. School of Medicine, Perdana University Royal College of Surgeons in Ireland (PURCSI), Kuala Lumpur 50490, Selangor, Malaysia
5. Faculty of Medicine, Biomedical Science and Nursing, MAHSA University, Jenjarom 42610, Selangor, Malaysia
6. Analytical Biochemistry Research Centre (ABrC), Universiti Sains Malaysia, Bayan Lepas 11900, Penang, Malaysia
* Correspondence: theingi@aimst.edu.my; Tel.: +60-1-6459-4792

**Abstract:** (1) Background: The teaching profession has become more challenging due to the increased use of information technology, which potentially increases psychological distress among teachers. This study aimed to determine the prevalence of depression, anxiety, and stress among school teachers in Malaysia during the period of the COVID-19 pandemic and its associated sociodemographic factors. (2) Methods: A cross-sectional study was conducted among primary and secondary school teachers in Malaysia. A validated DASS-21 questionnaire was used for mental health screening, and the study was conducted online during the pandemic period. (3) Results: The percentages of respondents with mild, moderate, severe, and extremely severe depression were 12%, 9.7%, 4.7%, and 3.1%, respectively. Those with mild, moderate, severe, and extremely severe anxiety accounted for 11.5%, 12.3%, 6.3%, and 6%, respectively. Those with mild, moderate, severe, and very severe stress accounted for 12.8%, 12%, 5.3%, and 2.5%, respectively. Perceived overworking was significantly greater during the pandemic compared to before the pandemic. Significant experience in teaching, and less perceived overworking before and during the pandemic were associated with better mental health. (4) Conclusions: Periodic mental health screening for teachers may be beneficial in preventing mental health disorders and improving the quality of student education. A full assessment and innovation of the curriculum and workload should be implemented.

**Keywords:** COVID-19 pandemic; teachers; depression; anxiety; stress

## 1. Introduction

The COVID-19 pandemic was one of the most devastating events. The virus infected thousands of people in day-to-day life in different age groups, including the working age group, and the global economy was greatly affected by the pandemic. [1]. The outbreak was first identified in Wuhan, China, in December 2019 [2,3]. The World Health Organization (WHO) declared the outbreak a Public Health Emergency of International Concern on 30 January 2020 and a pandemic on 11 March 2020 [4,5].

In response to the COVID-19 pandemic, the government of Malaysia and Ministry of Health (MOH) took action to manage disease spread and minimise mortality by implementing a movement control order (MCO) on 18 March 2020. The Director-General of the MOH emphasised that the order being enforced came under the Prevention and Control of Infectious Diseases Act 1988 and the Police Act 1967 and would help to control the spread of the virus. Public institutions, including educational institutions, schools, and higher education institutions, were required to close due to COVID-19 [6].

Teachers have been thrown a curveball by the COVID-19 pandemic and the closure of all schools across the country. Due to the COVID-19 epidemic, there is now a significant increase in the use of smartphones and other personal telecommunication devices for learning and teaching that were previously forbidden on school property. The traditional modes of teaching and learning, as well as educator duties, are impacted by the global crisis, which makes them more challenging. They must adapt to the new norm, which necessitates carrying out the procedure online. Teachers continue to impart knowledge via online platforms and social media, such as Google Classroom, Zoom, WhatsApp, and Telegram applications, during the movement control order (MCO) and the conditional movement control order (CMCO) to decrease the possibility of community transmission to ensure that no student falls behind and may continue learning in a secure way [7,8].

It is thought that teachers need to improve their knowledge, competency, and attitudes and be prepared to accept new norms related to teaching, especially online learning. However, not all teachers are comfortable with using digital technology, and especially older teachers do not easily adapt to the new classroom. Many senior teachers find it difficult to work around technology [7].

It is widely acknowledged that education is one of the most critical sectors with regard to work-related stress. Work-related stress in teachers was first identified in the 1930s by Smith and Milstein [9]. According to a 2005 comparative study by Johnson S. et al., after ambulance drivers, teaching is the profession that causes the most stress. [10].

According to a 2011 survey of 7853 teachers, Malaysian teachers work between 40 and 80 h per week, on average 57 h [11]. Burnout is known to be increased by emotional exhaustion, high demands, low job control, an excessive workload, and low reward [12]. Furthermore, personal characteristics, such as low self-efficacy and poor leadership, have been linked to burnout [13].

In the midst of the COVID-19 pandemic, health and mental health professionals faced challenges due to a lack of information about the psychological impact and underlying mental health conditions of the general public, especially teachers. Researchers are examining the virus's genome, the epidemiology and clinical traits of infected people, as well as the difficulties faced by healthcare administrators and functionaries. Researchers have looked into how the COVID-19 pandemic has affected students and healthcare professionals psychologically. In this COVID-19 pandemic, however, less attention is being paid to the impact of COVID-19 on the mental health of Malaysian school teachers.

Previous research has found that different levels of stress and factors are involved in different levels of the teacher population. Moreover, the teaching profession has recently undergone rapid development and has become more challenging due to the increasing use of information technology and globalisation in order to adapt to the new norm of conducting online teaching, which potentially increases the stress among teachers. As a result, the purpose of this study was to ascertain the incidence of psychological distress, such as depression, anxiety, and stress, among primary and secondary school teachers in Malaysia during the COVID-19 pandemic and its associated correlates.

## 2. Materials and Methods

Using an online questionnaire and convenience sampling, a cross-sectional study was conducted among Malaysian teachers during the COVID-19 pandemic from 1 August 2021 to 31 October 2021. The link to the questionnaire was sent through emails, WhatsApp, Telegram and other social media to the participants.

The online questionnaire contained sociodemographic details and the validated Depression, Anxiety, and Stress Scale (DASS) 21 in both Malay and English versions. DASS-21 consisted of 21 questions that were equally attributed to depression, anxiety, and stress. This 21-item, 4-point Likert scale was widely used to assess the negative emotional states of depression, anxiety, and stress over the past week. The assessment did not require special training and was suitable for non-clinical settings. A rating scale of (0) means that it "did not apply to me at all". A scale of (1) means that it "applied to me to some degree, or some

of the time". A scale of (2) means that it "applied to me to a considerable degree, or a good part of the time," while a scale of (3) means that it "applied to me very much, or most of the time" [14]. Teachers who worked in Malaysia during the period of the COVID-19 pandemic from 1 May 2021 to 1 May 2022 were included in the study. Those who worked as temporary replacement teachers in the schools or teachers who were on maternal leave or sick leave during the period of the COVID-19 were excluded from the current study.

The sample size calculation was carried out using a single proportion formula.

$$n = \frac{Z^2 \, P \, (1 - P)}{d^2}$$

- $n$ = sample size,
- $Z$ = Z statistic for a level of confidence,
- $P$ = expected prevalence or proportion (in proportion of one; if 20%, $P$ = 0.2),
- $d$ = precision (in proportion of one; if 5%, $d$ = 0.05).

The sample size was 345 participants to achieve 5% precision in estimating the prevalence of teachers with mental health problems via screening, which was 34% in 2009 using the Sample Size Calculator for Prevalence Studies (Naing et al., 2006) [15]. The total number of participants needed was 380 after taking into account a non-respondent rate of 10%.

The respondents were mainly from Perak and Penang states, which included both primary and secondary school teachers, teaching in urban and rural regions, regardless of whether they were working at private or government schools. The online questionnaire was randomly distributed to the respondents regardless of their age, gender, ethnicity, and religion. It was carried out as a preliminary study, mainly focusing in two states of Malaysia. The data analysis was carried out using SPSS version 22. The mean ± standard deviation (SD) was used to express the descriptive data. One-way ANOVA was used for the analysis of normally distributed variables. A Kruskal–Wallis ANOVA test was performed for the non-normally distributed data. Categorical data were analysed using Chi-square or Fisher 's exact test, and Pearson's correlation was used to test the association between the continuous variables. A value of $p < 0.05$ was considered statistically significant.

The study was registered in the National Medical Research Registry (NMRR-20-1700-55999). The approval to execute this study was obtained from the Medical Research Ethics Committee (MREC), Ministry of Health Malaysia. Informed consent was obtained from all participants via the online link.

## 3. Results

A total of 382 respondents were involved in this study and the response rate was 98%. The background socio-economic and educational characteristics of the respondents are demonstrated in Tables 1 and 2.

**Table 1.** Socio-demographic characteristics of the respondents (N = 382).

| Variables | Frequency | % |
|---|---|---|
| Age (years) | | |
| 18–30 | 86 | 22.5 |
| 31–40 | 130 | 34 |
| 41–50 | 96 | 25.1 |
| 51–60 | 68 | 17.8 |
| Older than 60 | 2 | 0.5 |
| Gender | | |
| Male | 64 | 16.8 |
| Female | 318 | 83.2 |
| Ethnicity | | |
| Malay | 92 | 24.1 |
| Chinese | 261 | 68.3 |
| Indian | 22 | 5.8 |

**Table 1.** *Cont.*

| Variables | Frequency | % |
|---|---|---|
| Other | 7 | 1.8 |
| Religion | | |
| Islam | 94 | 24.6 |
| Buddhism | 213 | 55.8 |
| Christian | 47 | 12.3 |
| Hinduism | 20 | 5.2 |
| Other | 8 | 2.1 |
| Marital status | | |
| Single | 136 | 35.6 |
| Married | 236 | 61.8 |
| Divorced | 5 | 1.3 |
| Widow(er) | 5 | 1.3 |
| Number of children | | |
| No child | 175 | 45.8 |
| One to two | 121 | 31.7 |
| Three to four | 72 | 18.8 |
| Five or more | 14 | 3.7 |
| Age of children | | |
| No child <14 y/o | 250 | 65.4 |
| 1–2 children <14 y/o | 102 | 26.7 |
| 3 or more children, 14 y/o | 30 | 7.9 |
| Family member | | |
| 0 to 2 | 82 | 21.5 |
| 3 to 5 | 214 | 56 |
| 6 to 8 | 73 | 19.1 |
| 9 or more | 13 | 3.4 |

**Table 2.** School and educational background of the respondents (N = 382).

| Variables | Frequency | % |
|---|---|---|
| Education | | |
| Diploma and equivalent | 30 | 7.9 |
| Degree | 296 | 77.5 |
| Master | 56 | 14.7 |
| Teaching experience | | |
| 0 to 5 years | 83 | 21.7 |
| 6 to 10 years | 65 | 17 |
| More than 10 years | 234 | 61.3 |
| Current teaching school | | |
| Primary | 241 | 63.1 |
| Secondary | 141 | 36.9 |
| Government or private | | |
| Government | 332 | 86.9 |
| Private | 50 | 13.1 |
| Location | | |
| Urban | 232 | 60.7 |
| Rural | 150 | 39.3 |
| Post holder | | |
| Principal/assistant | 64 | 16.8 |
| Class teacher | 172 | 45 |
| Subject teacher | 125 | 32.7 |
| Counselor | 9 | 2.4 |
| Other | 12 | 3.1 |
| Monthly household income | | |

**Table 2.** *Cont.*

| Variables | Frequency | % |
|---|---|---|
| Less than MYR 2000 | 9 | 2.4 |
| MYR 2000–4999 | 160 | 41.9 |
| MYR 5000–9999 | 186 | 48.7 |
| ≥MYR 10,000 | 27 | 7.1 |

Mean (SD) of teaching experience = 14.2 (9.1) years.

The paired *t* test in Table 3 showed no significant differences in sleeping or working hours before the pandemic or during the pandemic in different modes of teaching.

**Table 3.** Sleeping and working hours before and during the COVID-19 pandemic.

| Variables | Average | SD | Minimum | Maximum | t | *p* Value |
|---|---|---|---|---|---|---|
| Sleeping hours/day F2F teaching before pandemic | 6.8 | 1.1 | 3 | 12 | | |
| Online teaching during pandemic | 6.5 | 1.2 | 0 | 10 | 0.891 | 0.374 |
| Working hours/day F2F teaching during pandemic | 7.2 | 2.1 | 0 | 14 | | |
| Online teaching during pandemic | 7.6 | 3.4 | 0 | 20 | 0.934 | 0.351 |

The results in Table 4 show respondents with underlying diseases, pregnancy, and those who thought they overworked. Respondents' perception of being overworked was 40.6% before the pandemic and 67.8% during the pandemic. Figure 1 demonstrated the prevalence of different diseases among the teachers, showing 8.75% for hypertension, 7% for other diseases, 4.25% for asthma, 4% for diabetes, and 0.75% for heart disease.

**Table 4.** Underlying medical illness, pregnancy, and perception of being overworked (N = 382).

| Variables | Frequency | % |
|---|---|---|
| Medical illnesses | | |
| No illness | 301 | 75.3 |
| Diabetes | 16 | 4 |
| Hypertension | 35 | 8.8 |
| Asthma | 17 | 4.3 |
| Heart disease | 3 | 0.8 |
| Other disease | 28 | 7 |
| Pregnancy | | |
| Yes | 6 | 1.6 |
| No | 299 | 78.3 |
| Not applicable | 77 | 20.2 |
| Perceived to be overworked before pandemic | | |
| Yes | 155 | 40.6 |
| No | 227 | 59.4 |
| Perceived to be overworked during pandemic | | |
| Yes | 259 | 67.8 |
| No | 123 | 32.2 |

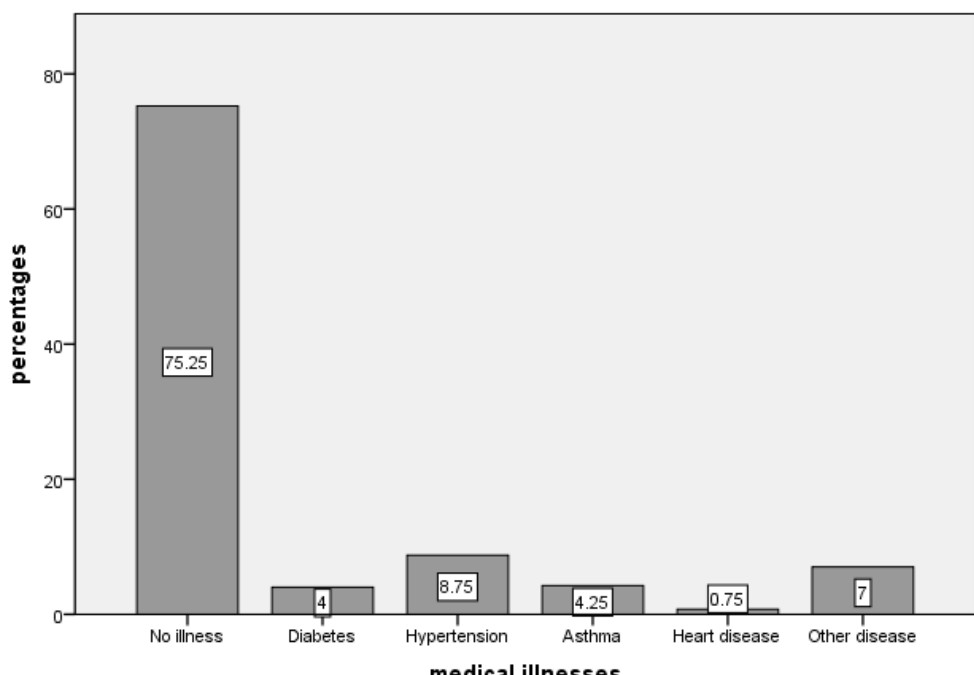

**Figure 1.** Underlying medical illnesses among the respondents.

In this study, those with mild depression accounted for 12% of respondents, moderate depression accounted for 9.7%, severe 4.7%, and extremely severe 3.1%. Those with mild anxiety accounted for 11.5%, moderate 12.3%, severe 6.3%, and extremely severe 6%. Those with mild stress accounted for 12.8%, moderate 12%, severe 5.3%, and extremely severe 2.5%. Gender was not statistically related to depression, anxiety, or stress in the current study (Table 5).

**Table 5.** Severity distribution of DASS scores among the respondents.

| Subscales | | Normal (%) | Mild (%) | Moderate (%) | Severe (%) | Extremely Severe (%) | $X^2$ | *p* Value |
|---|---|---|---|---|---|---|---|---|
| Depression | | | | | | | | |
| | All | 269 (70.4) | 46 (12) | 37 (9.7) | 18 (4.7) | 12 (3.1) | | |
| | Males | 44 (68.8) | 13 (20.3) | 4 (6.3) | 1 (1.6) | 2 (3.1) | 6.953 | 0.138 |
| | Females | 225 (70.8) | 33 (10.4) | 33 (10.4) | 17 (5.3) | 10 (3.1) | | |
| Anxiety | | | | | | | | |
| | All | 244 (63.9) | 44 (11.5) | 47 (12.3) | 24 (6.3) | 23 (6) | | |
| | Males | 40 (62.5) | 6 (9.4) | 14 (21.9) | 1 (1.6) | 3 (4.7) | 9.007 | 0.61 |
| | Females | 204 (64.2) | 38 (11.9) | 33 (10.4) | 23 (7.2) | 20 (6.3) | | |
| Stress | | | | | | | | |
| | All | 259 (67.8) | 49 (12.8) | 46 (12) | 20 (5.2) | 8 (2.1) | | |
| | Males | 49 (76.6) | 7 (10.9) | 5 (7.8) | 3 (4.7) | 0 (0) | 3.881 | 0.422 |
| | Females | 210 (66) | 42 (13.2) | 41 (12.9) | 17 (5.3) | 8 (2.5) | | |

Multiple linear regression analysis was carried out to identify the predictors of mental health in Table 6. In this study, teaching experience, the perception of being overworked before the COVID-19 pandemic and the perception of being overworked during the pandemic significantly affected depression, anxiety and stress of the respondents. Significant teaching experience and less perceived overworking before and during the COVID-19 pandemic were protective factors for depression, anxiety, and stress. Other sociodemographic factors did not depend on mental health based on the findings.

**Table 6.** Multiple Linear Regression predicting Depression, Anxiety and Stress.

| Predictor | Depression | | | | Anxiety | | | | Stress | | | |
|---|---|---|---|---|---|---|---|---|---|---|---|---|
| | t | β | Std. Error | *p* | t | β | Std. Error | *p* | t | β | Std. Error | *p* |
| Teaching experience | −4.395 | −0.209 | 0.022 | 0.000 * | −2.813 | −0.137 | 0.021 | 0.005 * | −3.165 | −0.149 | 0.023 | 0.000 * |
| Perceived to be overworked before pandemic | −2.495 | −0.123 | 0.418 | 0.013 * | −2.05 | −0.103 | 0.404 | 0.041 * | −2.516 | −0.123 | 0.443 | 0.012 * |
| Perceived to be overworked during pandemic | −5.919 | −0.289 | 0.436 | 0.000 * | −5.58 | −0.279 | 0.421 | 0.000 * | −7.11 | −0.343 | 0.461 | 0.000 * |

\* $p < 0.05$, statistically significant.

Table 7 shows the association between DASS score and sleeping and working hours among the respondents. Positive correlation was noticed between working hours per day for online teaching during the pandemic and the respondent's depression and stress. The strength of associations was weak. In this study, the longer the working hours for online teaching, the more depression and stress the respondents experienced.

**Table 7.** Association of DASS scores with sleeping and working hours of the respondent.

| Variables | Categories(n) | Depression | Anxiety | Stress |
|---|---|---|---|---|
| Sleeping hours per day when conducting F2F teaching before pandemic | Continuous variable | r = −0.058 p = 0.256 | r = −0.061 p = 0.237 | r = −0.040 p = 0.431 |
| Sleeping hours per day when conducting online teaching during pandemic | Continuous variable | r = −0.033 p = 0.518 | r = −0.031 p = 0.545 | r = −0.027 p = 0.599 |
| Working hours per day when conducting F2F teaching during pandemic | Continuous variable | r = −0.030 p = 0.560 | r = −0.065 p = 0.205 | r = 0.070 p = 0.172 |
| Working hours per day when conducting online teaching during pandemic | Continuous variable | r = 0.113 p = 0.028 * | r = 0.100 p = 0.051 | r = 0.150 p = 0.003 ** |

\* $p < 0.05$, \*\* $p < 0.01$, statistically significant.

## 4. Discussion

In the present study, no significant differences in sleeping and working hours were found before the pandemic or during the pandemic in either face-to-face or online teaching (Table 3). According to Marelli S et al., (2021), prior to and during the COVID-19 emergency, there was an increase in bedtime hour, sleep latency, and wake-up time, as well as a worsening of sleep quality and insomnia symptoms [16]. The impact of the delay in bedtime and wake-up was most noticeable in students during the lockdown. Lockdown following the COVID-19 pandemic has significantly altered students' lifestyles, transforming them into sedentary lifestyles. Students experienced increased screen time due to online teaching or assignments, as well as changing sleep durations and patterns. The majority of students went to bed between 23 and 24 P.M., while the majority of students awoke at 8 A.M. [16]. According to Nor Asma Musa et al., (2018), total teaching hours, depression, and stress were significantly associated with poor sleep quality [17].

Respondents who thought that they were overworked accounted for 40.6% before the pandemic and 67.8% during the pandemic. Psychological distress among teachers is predicted by job demands, which is in line with previous findings [18,19]. Siti Rapidah Omar Ali et al. (2017) mentioned that workload pressure is also defined as a shift in the level of stress experienced by employees, which has an impact on their performance. Job stress and a heavy workload can have a detrimental impact on employees' performance and increase the risk of occupational health hazards [20]. Before the pandemic, teachers were required to run administrative duties, such as programmes and enter data for online

evaluation requirements even after midnight to avoid system overload during working hours, in addition to their routine teaching duties [20]. During the pandemic, teachers were further burdened with substantial online teaching tasks and monitoring of students' progression. Longer working hours for online teaching instead of face-to-face teaching contributed to depression and stress for our participants. A US study indicated higher screen time, such as more than six hours of computer use per day, was associated with depression [21]. E-mail overuse for monitoring students' homework and conveying messages to parents or students may be a contributing factor of burnout for teachers [22]. Significant experience in teaching and less perceived overworking before and during the COVID-19 pandemic were protective factors for depression, anxiety and stress (Table 6). In this study, the longer the working hours for online teaching, the more depression and stress respondents experienced (Table 7). The above studies are matched with the present survey.

The prevalence of different diseases among the teachers showed 8.75% for hypertension, 7% for other diseases, 4.25% for asthma, 4% for diabetes, and 0.75% for heart disease (Table 4), which coincides with the work of the mentioned authors. Wang Y, Li Q, Tarimo CS, et al. (2021) reported that all age-group teachers were "very concerned" about the COVID-19 outbreak. Female teachers outnumbered male teachers in the classroom [23]. Adam W. Gaffney, David Himmelstein, and Steffie Woolhandler (2020) mentioned that some teachers had definite or possible risk factors for severe COVID-19 illness, such as cancer, a higher body mass index (BMI), and cardiac conditions, such as like disease and diabetes, including 2.50 million teachers who were older than 64 years and 4.67 million with heart disease [24].

Those with mild depression accounted for 12% of the respondents, moderate 9.7%, severe 4.7%, and extremely severe 3.1%. Those with mild anxiety accounted for 11.5%, moderate 12.3%, severe 6.3%, and extremely severe 6%. Those with mild stress accounted for 12.8%, moderate 12%, severe 5.3%, and extremely severe 2.5%. Gender was not statistically related to depression, anxiety, or stress in the current study (Table 5). The present study noticed that one in three to four teachers experienced depression, anxiety, and stress during the COVID-19 pandemic in our study. The prevalence of stress was consistent with a meta-analysis that encompassed primary, secondary, and tertiary teachers, but the alarming prevalence of depression and anxiety among our teachers was higher compared to the meta-analysis [25].

This study is limited by its cross-sectional nature and convenient sampling, as the sample is small and not countrywide, with limited generalisability and a lack of representativeness. The findings may not be generalised to teachers in other geographical locations, especially in the east of Malaysia, as there may be unmeasured confounding variables in this study. Despite these limitations, this study provides a reasonably accurate preliminary study to determine the prevalence of depression, anxiety, and stress among school teachers during the period of the COVID-19 pandemic. A larger sample size study, which is more representative, will be carried out in future studies based on the findings from the current study. The current studies highlight that suitable working hours and proper training related to online, as well as blended teaching, for teachers are required for a healthy working environment.

## 5. Conclusions

It is critical to manage stress, but job security and family responsibilities, particularly for less experienced teachers, are major concerns during the pandemic. Findings from this study suggest that one should pay more attention to vulnerable groups, such as teachers with less experience and those who perceive themselves to be overworked. Nationwide strategic plans, such as psychological first aid during major disasters and a comprehensive review and improvisation of the hybrid curriculum and workload, should be established. Periodically, mental health screening for teachers may be useful to prevent mental health problems and improve the quality of education for students.

**Author Contributions:** Conceptualization, S.Y.T. and M.S.K.; methodology, S.Y.T. and C.L.T.; software, T.M.M.; validation, C.L.T., T.M.M. and S.Y.T.; formal analysis, T.M.M.; writing—original draft preparation, S.Y.T. and C.L.T.; writing—review and editing, L.S. and T.Y.C. All authors have read and agreed to the published version of the manuscript.

**Funding:** This research received no external funding.

**Institutional Review Board Statement:** The study was registered in the National Medical Research Registry (NMRR-20-1700-55999). The approval to execute this study was obtained from the Medical Research Ethics Committee (MREC), Ministry of Health Malaysia (effective date 25 August 2021). The study was conducted in accordance with the Declaration of Helsinki.

**Informed Consent Statement:** Informed consent was obtained from all subjects involved in the study.

**Data Availability Statement:** Not applicable.

**Acknowledgments:** We would like to thank our Director-General of Health, Malaysia, Noor Hisham Abdullah for his permission to conduct this study. We would like to thank all respondents from the primary and secondary schools for taking time to answer the questionnaire.

**Conflicts of Interest:** The authors declare no conflict of interest.

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
