# Peer review of "Mental Health Screening during COVID-19 Pandemic among School Teachers in Malaysia: A Cross-Sectional Study"

_sustainability, doi:10.3390/su141710664_

Round 1
Author Response
Changes had been made based on the reviewer's feedback. Please see the attachment.

Reviewer 2 Report
The authors conducted a cross-sectional study to examine the prevalence of depression, anxiety, and feelings of stress among teachers in Malaysia during the COVID-19 pandemic.
Since teaching staff had to compensate lockdown measures with online teaching and digital media use, the question of increased stress with mental health consequences is quite relevant.
However, there are a number of limitations of the methodology and ambiguities in the reporting of the study. I would like to present these in the following.
Introduction
The introduction is clearly written and the research question clearly derived at the end. At random, I have read the cited publication [12]. The study reported here has by no means 7,853 participants, but 356, so is subject to the same limitations as the study reported in the manuscript (see below). Othman et al are misquoted here, which needs to be corrected.
Material and Methods
It remains unclear how large the universe of eligible participants is. Given a population size of 32 million in Malaysia, I would estimate the number of teachers to be at least 200,000. It is also unclear whether only teachers from the primary sector or also the secondary sector were invited, as well as whether all states were involved.
The case number estimate is plausible with respect to the outcome, but a non-responder rate of 10% is very unlikely and does not correspond to experience in cross-sectional studies.
Results
The number of participants of 382 is extremely small and not understandable in light of the totality of teachers in Malaysia. The lack of representativeness can also be seen in the reported number of Chinese and Buddhists, 68% Chinese in the study, 23.7% in the normal population, 55.8% Buddhists in the study, 19.2% in the normal population. Therefore, the authors should be very transparent about the study sample here. It must be visible how many of the participants who could have participated did not and what the response-rate actually was.
The authors report a large number of results from individual statistical tests. These include confounders that are not independent of each other. For example, it is to be expected that higher age may be related to the number of children. Certainly, the number of family members is associated with the number of children. Therefore, it would be important to finally examine all confounders considered relevant in a joint multivariate model to determine what remains significant in the end.
Discussion
The discussion should start with a short summary of the results. Only after that should a comparison with the literature be made (sentence about meta-analysis should be moved to the bottom).
The limitations are completely missing, the lack of representativeness, the unknown non-responder rate, the very optimistic case number estimation. The applicability of the results to the total population of teachers needs to be discussed before conclusions can be drawn for preventive measures.
All sorts of spelling or formatting errors can still be found.
Author Response
The corrections have been made in the manuscript. Please see the attachment for details.

Round 2
Reviewer 2 Report
The authors have substantially revised their manuscript.
However, some of my comments have not been addressed.
Introduction
The reference of Othman et al has now been removed and replaced by another study. However, the link provided is incorrect and the source is not accessible below.
Material and methods
It remains unclear how large the total eligible participants are. With a population size of 32 million, I would estimate the number of teachers to be at least 200,000. There is now talk of a convenience sample, but it remains unclear where the addresses for the emails, WhatsApp and Telegram invitations came from. How many people were contacted in total?
Results
It remains unclear how many of the participants who could have taken part did not and how high the response rate actually was.
There are still spelling and grammatical errors.
Author Response
Thank you so much for your comments. We have responded to the area concerned in the attachment.
